# The Genomic Evolution and the Transmission Dynamics of H6N2 Avian Influenza A Viruses in Southern China

**DOI:** 10.3390/v14061154

**Published:** 2022-05-26

**Authors:** Zhaoxia Yuan, Taifang Zhou, Jiahao Zhang, Qingxin Zeng, Danli Jiang, Meifang Wei, Xudong Li

**Affiliations:** 1College of Animal Science & Technology, Zhongkai University of Agriculture and Engineering, Guangzhou 510642, China; zengqingxin1995@163.com (Q.Z.); danli0222@163.com (D.J.); 18826234364@163.com (M.W.); 2Guangdong Province Key Laboratory of Waterfowl Healthy Breeding, Guangzhou 510225, China; haoszhang1994@126.com; 3College of Veterinary Medicine, South China Agricultural University, Guangzhou 510642, China; zhoutaifang@163.com (T.Z.); lxd6888@126.com (X.L.); 4National Avian Influenza Para-Reference Laboratory, Guangzhou 510642, China; 5National and Regional Joint Engineering Laboratory for Medicament of Zoonoses Prevention and Control, National Development and Reform Commission of People’s Republic of China, Guangzhou 510642, China

**Keywords:** avian influenza virus, H6 subtype virus, evolution, transmission dynamics

## Abstract

In China, the broad prevalence of H6 subtype influenza viruses, increasingly detected in aquatic birds, promotes their exchange materials with other highly pathogenic human-infecting H5N1, H5N6, and H7N9 influenza viruses. Strikingly, some H6 subtype viruses can infect pigs, dogs, and humans, posing risks to public health. In this study, 9 H6N2 viruses recovered from waterfowl species in the Guangdong province of China in 2018 were isolated and sequenced. Phylogenetic analysis revealed that the genome sequences of these H6N2 viruses belonged to Group I, except for the NP gene in Group III. Coalescent analyses demonstrated that the reassortment of NA and NS genes have occurred in two independent clusters, suggesting H6 subtype viruses had been undergoing a complex reassortant. To examine the evolutionary dynamics and the dissemination of the H6 subtype viruses, a Bayesian stochastic search variable selection was performed for results showing higher viral migration rates between closer provinces, including Guangdong, Jiangxi, Guangxi, and Fujian. Notably, the transmission routes of the H6 subtype viruses were concentrated in Jiangxi Province, the most frequent location for input and output transmission and a region containing Poyang Lake, a well-known wintering site for migration birds. We also found that the aquatic birds, especially ducks, were the most common input source of the viral transmission. In addition, we also found that eight positively selected amino acid sites were identified in HA protein. Given their continuous dissemination and the broad prevalence of the H6 subtype influenza viruses, continued surveillance is warranted in the future.

## 1. Introduction

Influenza A viruses not only cause severe economic damage to poultry but also pose major threats to human health. Based on the antigenic difference of surface protein in the hemagglutinin (HA) and neuraminidase (NA), the influenza A virus can be, respectively, categorized as either H1–16 and N1–9 or H17–18 and N10–11, the latter two of which had been detected only in bats [1]. Emerging in diverse combinations of HA and NA, avian influenza viruses (AIVs) can form many subtypes mainly including H5N1, H5N2, H5N6, H5N8, H7N9, and H9N2. Aquatic birds are major reservoirs of AIVs, which spread rapidly among wild birds [2,3].

The population of domestic ducks has increased significantly in China in the last 20 years. As the site where a whopping 75% of domestic ducks are bred across the globe, China is regarded as the largest reservoir for influenza viruses carried by aquatic birds worldwide [4,5]. In recent decades, systematic surveillance has shown that H5N1, H5N6, and H7N9 HPAIVs have emerged on a large scale in China and, in the process, wreaked havoc on poultry industry and public health. Perhaps more concerning, since 2003, 861 cases of human infection with H5N1 AIVs have been confirmed, 455 of which ended in death, for a case fatality rate of approximately 53% [6]; and, since 2013, 1567 cases of infection with H7N9 AIVs have been confirmed, 615 of which were fatal, for a case fatality rate of approximately 39% [7], posing considerable threats to public health. Relative to HPAIVs, low pathogenic AIVs (LPAIV) cause only mild clinical disease in poultry, commonly associated with a decline in egg production, abnormal eggs, and high morbidity but low mortality [8,9]. Of particular note is that the H6 subtype viruses in aquatic birds rose significantly in China in recent years.

H6 subtype viruses were first isolated in a turkey in Massachusetts in 1965 and has become prevalent worldwide over the past decade, causing huge losses to the poultry industry [10,11,12,13]. More recently, H6 subtype viruses have been found to cause largely asymptomatic infections in waterfowl [14,15], which gives them a more extensive range of host species than all other subtypes [16]. From a phylogenetic standpoint, H6 subtype viruses can be classified as having two lineages—North American and Eurasian [5]—the latter of which primarily infects aquatic birds. By extension, the viruses of Eurasian lineage form three major groups—Group I (i.e., ST339-like), Group II (i.e., ST2853-like), and Group III (i.e., HN573-like)—with H6Nx viruses in Groups I and II currently co-circulating in China [10,13].

Over time, H6 subtype viruses have begun to adapt to mammalian species and, in turn, to provide internal genes for the reassortment of HPAIVs. Research has shown that 34% of H6 subtype AIVs have been able to simultaneously bind to human receptors [13]. Some H6 subtype viruses can infect and cause disease in mice and ferrets [17,18] and, in 2010, H6N6 viruses were detected in pigs in the Guangdong Province of China [19]. Three years later, a case of human infection with an H6N1 virus involving flu-like symptoms was reported in Taiwan [20]. In addition, a study on seroprevalence showed that H6 subtype viruses had seropositivity among occupational contact workers in 19 provinces in China [21]. For our part, we recently demonstrated that reassorting human- and bird-origin H5N6 viruses all harbored a segment of H6 origin [22,23]. Altogether, those findings indicated that H6 subtype viruses pose a significant threat to public health.

In southern China, which is known for facilitating the emergence and the transmission of pandemic influenza viruses, poultry farming occurs in high-density environments and with free-range techniques [13,24]. Live poultry markets in China frequently allow close contact between migratory birds and poultry, promoting the rapid spread of AIVs and frequent genetic reassortment, in addition to a culture of poultry breeding without any biosecurity measures that foster environments for birds to share the water, food, and habitats. However, despite phylogenetic analyses of H6 subtype viruses, relevant research on the evolution and the transmission dynamics in China has been slight. In response, we fully characterized the genetic evolution and the transmission dynamics of H6 subtype viruses, particularly H6N2 viruses, to clarify evolutionary processes through which they circulate in China.

## 2. Material and Methods

### 2.1. Sample Collection and Virus Isolation

From January to December 2018, we collected 1348 cloacal and tracheal swab samples of chickens, ducks, and geese from live poultry markets in Guangdong, China. All animals involved in experiments were reviewed and approved by the Institution Animal Care and Use Committee at South China Agricultural University, and they were treated in accordance with the guidelines (2017A002). Each sample was placed in 2 mL of the PBS supplemented with penicillin (5000 U/mL) and streptomycin (5000 U/mL). All the samples were inoculated in the allantoic cavities of 10-day-old embryonated chicken egg at 37 °C. The allantoic fluid was collected and then tested via HA assay with 1% chicken red blood cells before use.

### 2.2. RNA Extraction, RT-PCR and DNA Sequencing

Before amplication of the full-length genome sequences, virus isolation from the swab samples were conducted and the HA and the NA genes of positive samples were firstly amplified and identified. Then, RNA was extracted from the suspension of 9 H6N2 influenza viruses with the RNeasy Mini Kit available from QIAGEN (Guangdong, China). A two-step RT-PCR was conducted with universal primers as previously described [25]. Eight gene sequences of H6N2 viruses were amplified using PrimeSTAR Max DNA Polymerase (TAKARA) with frame-specific primers. PCR products were purified with a Gel Extraction Kit D2500 (OMEGA) and the gene sequences were sequenced by TSINGKE Co., Ltd. (Guangdong, China). The detailed information of gene sequences is available from Appendix A.

### 2.3. Phylogenetic Analysis

All available full genome sequences with the complete coding regions of H6 subtype viruses were downloaded from GenBank and GISAID. The number of reference sequences of PB2, PB1, PA, HA, NP, NA, M, and NS were 568, 527, 558, 532, 570, 411, 499, and 425, respectively (accessed 18 September 2020). Once the dataset of full genome sequences was created (alignment of sequences available upon request), the downloaded sequences together with 10 H6N2 strains were aligned in MAFFT (version 7.149) [26]. The maximum likelihood (ML) phylogenies for the codon alignment of eight gene segments were estimated using the GTR+G nucleotide substitution model in the RAxML (version 8.2) program [27]. Node support was determined by nonparametric bootstrapping with 1000 replicates, after which the phylogenetic tree was visualized in FigTree (version 1.4.3; http://tree.bio.ed.ac.uk/software/figtree/) (accessed on 8 December 2019).

### 2.4. Bayesian Maximum Clade Credibility Phylogeny and Evolutionary Dynamics Analysis of H6 Subtype Influenza Viruses

The package BEAST package (version 1.8.2) was used to estimate the time to the most recent common ancestor (tMRCA) and the evolutionary rates [28,29]. We estimated rates of evolutionary change (i.e., nucleotide substitution) in the HA and the NA gene segments of H6N2 viruses together with other H6 subtype viruses circulating in China. For efficiency’s sake, we focused on H6 subtype viruses sampled at different times and locations to reflect the representative phylogenetic diversity of the viruses. To ensure that the viruses had a temporal structure (i.e., in terms of HA and NA alignment) sufficient to allow the reliable estimation of rates, we performed a regression of root-to-tip genetic distances in the ML tree against exact sampling dates using TempEst, which involved removing anomalous sequences based on the estimated distances.

To obtain more robust estimated rates, we used the Bayesian Markov chain Monte Carlo (MCMC) method implemented in BEAST (version 1.8.2), which entailed using the GTR nucleotide substitution model, an uncorrected lognormal (UCLN) relaxed molecular clock model and a model of the Gaussian Markov random field (GMRF) Bayesian skyRide coalescent tree prior. Two runs of the MCMC method were combined using LogCombiner (version 1.8.3), with 100,000,000 total steps for each set and sampling every 10,000 steps, and the convergence (i.e., effective sample sizes > 200) of relevant parameters was assessed using Tracer (version 1.6; http://beast.bio.ed.ac.uk/) (accessed on 7 December 2019). The posterior distribution of trees obtained from BEAST analysis, with 10% of runs removed as burn-in runs, was also used to obtain the Bayesian maximum clade credibility tree for the HA and the NA gene segments.

Next, to estimate the population dynamics of H6N2 and H6N6 viruses in China, we inferred the demographic history of the viruses using Bayesian skyline plots. First, we performed a regression of root-to-tip genetic distances on the ML tree against exact sampling dates using TempEst [30]. Second, we performed BEAST analysis on the HA gene segments of H6N2 and H6N6 viruses in China, with the same parameters for estimating relative genetic diversity.

### 2.5. Phylogeographic Analysis of H6 Subtype Influenza Viruses

We accessed 868 HA genome sequences of H6 subtype viruses circulating in China from the GISAID Database. For efficiency’s sake, once duplicate entries and recombinant strains identified using RDP4 were removed, we focused on H6 subtype viruses sampled at different times and locations. In turn, identical sequences from the same year and region were removed. To mitigate potential sampling biases, we randomly subsampled the database in a stratified way to create a more equitable spatiotemporal distribution of the HA genome sequences, and time-measured phylogenies were inferred using a Bayesian discrete phylogeographic approach in BEAST (version 1.8.2). In that process, we first performed a regression of root-to-tip genetic distances in the ML tree against exact sampling dates using TempEst, which revealed a strong temporal signal. Second, we used an UCLN relaxed molecular clock model for GTR nucleotide substitution in BEAST, along with a Bayesian stochastic search variable selection (BSSVS) model with asymmetric substitution, to perform the phylogeographic analysis. For each independent dataset, multiple runs of the MCMC method were combined using LogCombiner (version 1.8.3), with 100,000,000 total steps for each set and sampling every 10,000 steps.

Next, to develop interactive visualizations of the dispersal process through time, we used SpreaD3 (version 0.9.6), which takes a rate matrix file for location states generated from analysis in BEAST using BSSVS [31]. We also use SpreaD3 to compute a Bayes factors (BF) test able to assess support for significant individual transitions between distinct geographic locations. On the test, BF values exceeding 100 indicate robust statistical support, from 30 to 100 indicate very strong statistical support, from 10 to 30 indicate strong statistical support, from 3 to 10 indicate substantial statistical support, and less than 3 indicate poor statistical support. We used R (https://www.r-project.org) (accessed on 3 January 2020) and ArcGIS 10.4 for Desktop (http://www.esri.com/software/arcgis/arcgis-for-desktop/) (accessed on 13 January 2020) to plot the results of the BF test.

### 2.6. Positive Selection Analysis

Selection pressures acting on the HA genes of H6Nx were analyzed by calculating the ratio non-synonymous to synonymous (dN/dS ration, *ω*) using the HyPhy software package and using Datamonkey (http://www.datamonkey.org/) (accessed on 10 March 2020). Selection pressures were evaluated according to the value of *ω* (i.e., *ω* < 1: negative selection, *ω* = 1: neutrality, *ω* > 1: positive selection). Four algorithms—single-likelihood ancestor counting (SLAC), fast unconstrained Bayesian approximation (FUBAR), mixed effects model of evolution (MEME), and fixed effects likelihood (FEL)—were used to estimate the positive amino acids sites of HA protein. Codons were considered to be under selection if confirmed by three of the algorithms, and the amino acid sites were considered to be under positive selection if confirmed by at least two of them (i.e., *p* < 0.1 in SLAC, *p* < 0.1 in FEL and MEME and *p* > 0.9 in FUBAR).

## 3. Results

### 3.1. Distribution and Evolution of H6 Subtype Viruses

From January to December 2018, we collected 1348 cloacal and tracheal swab samples of chickens, ducks, and geese from live poultry markets in Guangdong, China. A total of 9 H6N2 viruses were isolated during this period. We systematically analyzed the prevalence of H6 subtypes in China according to the time and the location distribution. Our findings suggested that the H6N1, H6N2, and H6N6 viruses were the main H6 subtypes in China (Figure 1A). From 2000 to 2005, H6N2 viruses were widespread in China; however, after 2005, the H6N6 had become the dominant H6 subtype in China and the number of H6N1 viruses began to decrease rapidly (Figure 1B). We also found that H6N2 viruses were mainly distributed in southern China, such as Guangdong Province; however, in northern China, we found that the numbers of H6N6 isolates were higher than other H6 subtype viruses and the H6N3, H6N4, H6N5, and H6N8 viruses were found in Henan, Gansu, and HongKong, China (Figure 1B).

To elucidate the evolutionary process of H6N2 influenza viruses in southern China, we examined 9 full genomes of new H6N2 viruses isolated in the region in 2018, together with other H6 subtype viruses, by performing multiple sequence alignment and phylogenetic analysis. In a previous study [10], we categorized H6 subtype viruses into three groups—Group I (e.g., ST339-like viruses), Group II (e.g., ST2853-like viruses), and Group III (e.g., HN573-like viruses)—as shown in Figure 2. Although the surface genes of H6N2 viruses were clustered into Group I lineage, the internal genes of H6N2 viruses had clearly undergone a complex reassortment (Appendix A). Meanwhile, the PB1 and NP genes of H6N2 viruses had clustered together and belonged to the Group III lineage. The HA genes of phylogenetic trees indicated that H6 viruses in Group III contained higher genetic diversity of NA subtype combinations, including N1, N2, N4, N5, N6, and N8 (Figure 2), and all of the HA genes of our H6N2 isolates clustered together. The NA and the NS genes of H6N2 viruses were classified in Group I, and they had completely bifurcated into two independent branches (Figure 2; Appendix A) that originated from H6N2 viruses in the Guangdong and the Guangxi provinces of China, thus showing that the H6N2 strains circulating in southern China had experienced distinct reassortment events in different periods. In addition, we found 15 and 20 amino acid differences in the NA and the NS1 proteins of H6N2 viruses in two different branches, suggestive of the molecular differences of these H6N2 viruses. It is noteworthy that some H6N2 viruses isolated from 2014 were divided into the H9N2 gene pool, indicating that frequent reassortment of H6N2 and H9N2 viruses occurred in aquatic bird-origin influenza viruses.

### 3.2. Evolutionary Dynamics of H6 Subtype Viruses

Using a root-to-tip regression, our analysis of temporal structure revealed aspects of the clock-like structure of HA (*n* = 297, correlation coefficient = 0.72; *R*^2^ = 0.51) and NA (*n* = 386, correlation coefficient = 0.74; *R*^2^ = 0.55). Once the epidemic H6Nx viruses were classified to Groups I–III, we estimated the time of origin of each H6Nx virus in each group with 95% highest probability density as follows: Group I (1993.067–1997.089), Group II (1992.472–1999.561) and Group III (1998.228–2000.860) for HA (Figure 3A) and Group I (1984.523–1986.754), Group II (1986.053–1993.476) and Group III (1990.577–1995.030) for NA (Figure 3B). Among the results, although the tMRCAs of the HA genes in Group I and Group II were earlier than their counterparts in Group III, in the NA genes we observed that the tMRCAs in Group III were later than those in Group I and in Group II. However, there were no correlations in the evolutionary process of H6 subtype viruses in three independent lineages. We also demonstrated that both HA and NA genes in the H6N2 isolates from Guangdong Province belonged to Group I. The HA genes of those isolates formed a single cluster, and they showed a tMRCA of April 2018 (Figure 3A), whereas the NA genes, despite exhibiting the same tMRCA, formed two independent clusters (Figure 3B).

To compare the genetic diversity of the two major H6 subtype viruses in China—that is, H6N2 and H6N6—we inferred the HA genes of their viral demographic history using the Bayesian skyline plots. Our findings suggest that the effective population size of H6N2 viruses fluctuated only slightly until early 2013, when it deceased sharply until 2018 (Figure 4D). By contrast, the effective population size of H6N6 viruses showed a sharp increase from early 2004 to early 2005 and stably maintained until early 2006; followed by a slow decrease until mid-2007. A second sharp increase occurred during late 2008 to early 2009 and then stably maintained, indicating that the effective population size of H6N2 and H6N6 viruses differed noticeably (Figure 4E). According to those findings, the effective population size of H6N6 viruses was far higher than that of H6N2 viruses.

### 3.3. Spread of H6 Subtype Viruses

We also reconstructed the spatial dispersal networks of H6Nx viruses in China involving transmission routes, with BF values exceeding three and encompassing nine discrete viral sampling locations. Among the results, 25 significant transmission routes of H6Nx viruses were observed, with Guangdong, Jiangxi, and Hunan Provinces acting as epicenters for their spread (Figure 4A and Figure 5). Whereas Guangdong was linked with four locations—Guangxi (BF = 24,441), Fujian (BF = 48,892), Zhejiang (BF = 9), and Jiangxi (BF = 6)—Jiangxi was linked with eight: Zhejiang (BF = 1348), Hunan (BF = 283), Hubei (BF = 142), Guangxi (BF = 137), Guangdong (BF = 65), Guizhou (BF = 16), Henan (BF = 11) and Jiangsu (BF = 7). Beyond that, Hunan was linked with five locations—Jiangxi (BF = 6510), Guizhou (BF = 32), Jiangsu (BF = 9), Yunnan (BF = 4), and Henan (BF = 4)—whereas Guangxi was linked with only two: Guangdong (BF = 31) and Hubei (BF = 6) (Table 1). We additionally observed that higher viral migration rates had arisen between closer provinces, including from Hunan to Jiangxi (migration rate = 2.69), from Jiangxi to Hunan (migration rate = 2.11), from Hunan to Guizhou (migration rate = 1.67), and from Jiangxi to Zhejiang (migration rate = 1.53) (Table 1). Those findings suggest that Jiangxi Province, contributed the most to the viral output and input during the period. As shown in Figure 4C, the co-circulation of viral spread between Hunan and Jiangxi provinces and between Hubei and Jiangxi provinces was also observed using the phylogeographic approach. However, we found no significant correlation between the migration rates and the distance of the sampling locations (Figure 4B).

### 3.4. Host Transition of H6 Subtype Viruses

To estimate the transmission of H6Nx viruses in different poultry species in China, we applied a BSSVS procedure to transmission data for five chickens, ducks, geese, quails and wild birds. Statistical support was strong for viral migration from ducks to geese (BF = 741), from ducks to chickens (BF = 223), and from wild birds to geese (BF = 110). By contrast, support for viral migration from geese to chickens (BF = 14) and from quails to ducks (BF = 17) was rather weak (Figure 6), and it does not justify inferring the transmission of H6Nx viruses between those hosts. However, we can infer that aquatic birds, especially ducks, represented the most common input source for the transmission of H6Nx viruses, as consistent with previous findings demonstrating that H6 subtype viruses are most frequently encountered in aquatic birds.

### 3.5. Molecular Characterization

All H6N2 viruses isolated in our study exhibited the sequence motif PQIETR↓GLF, which had only one basic amino acid, arginine, in the cleavage site between HA1 and HA2 (Table 2), and that cleavage sequence was usually found in H6 viruses from terrestrial poultry. According to our data, none of the isolates possessed a sequence with multiple basic amino acids, which thus fulfilled the characteristics of LPAIVs. In HA receptor-binding sites, changes in amino acids such as A138S, Q226L, and G228S, despite having been reported to increase affinity for human-type receptors [32,33], were not detected in those positions in our H6N2 isolates (Table 2). However, the G228S substitution was found in H6 isolates from both pigs and humans, thereby indicating that H6N2 viruses may be able to infect mammals and humans. The human-infecting H6N1 virus had a 27-nucleotide deletion in the NA stalk region, with nine amino acids at positions 60 to 68 (Table 2), which may be associated with increased virulence in mammals. However, in our H6N2 isolates, the deletion in that region was not observed, nor were E119V, H275Y, R293K, and N295S substitutions in the NA protein, suggesting that these isolates were sensitive to neuraminidase inhibitors (e.g., Oseltamivir). Last, no S31N substitutions causing amantadine resistance were found in the M2 protein and no amino acid mutations were associated with receptor-binding affinity in the HA protein, but virulence in mammals was detected in the PB2, PA, and NS proteins (Table 2).

### 3.6. Positive Selection Analysis of H6 Subtype Viruses

Eight residues—in H6 numbering 4, 139, 145, 156, 170, 174, 300, and 388—in the HA protein of H6Nx viruses exhibited a positive selection (Figure 7). Residues 4, 139, 145, 156, 170, 300, and 388 in the protein were under positive selection according to SLAC, FUBAR, MEME, and FEL whereas SLAC, MEME, and FEL confirmed that residue 174 had undergone positive selection (Figure 7). All of these residues were polymorphic, indicating that these positively selected sites had been undergoing continuous evolution. Except for residue 388, all positively selected sites were located at the head of the HA1 subunit, which suggests high variations in HA1. Although no positively selected sites emerged in the antigenic sites A, C, D, and E, positively selected site 174 surfaced in antigenic site B. Thus, antigenic site B, closest to the receptor-binding site, is proposed to contribute to antigenic changes in the HA protein. In addition, the A139T, S145R, E174A, and K388R substitutions of HA protein were observed in H6 viruses (Figure 7). However, the biological function requires further investigation. It is noteworthy that A139T and E174A mutations were observed in all H6N2 isolates, indicating that H6 subtype viruses were continuously undergoing adaptive evolution.

## 4. Discussion

Detected with increasing frequency worldwide, H6 subtype influenza viruses posed significant threats to both poultry and human health. In several studies, multiple H6 subtype viruses have been detected in aquatic birds and terrestrial poultry in China [13,34,35], and given their broad prevalence, they continue to share genetic materials with other highly prevalent H5N1, H5N6, and H7N9 influenza viruses [36,37,38]. Other research showing the preference for human receptors of the human-infecting H6N1 virus posed a threat to human health [39].

Our analytical results showed that since 2005, the H6N2 and the H6N6 subtypes had replaced the H6N1 subtype as the dominant subtype in China. In addition, the collection sites of H6 isolates were mainly distributed in southern China. Recent studies had also demonstrated that H6N2 and H6N6 viruses had been co-circulating in China, especially in live poultry markets [10,13,40], while others have confirmed that H6N6 has supplanted H6N2 as the dominant H6 subtype since 2012 [34,41]. In our study, we observed that the genetic diversity of H6N2 viruses fluctuated slightly, followed by a sharp decrease, whereas the genetic diversity of H6N6 viruses, after decreasing slowly from 2006 to 2008, subsequently increased and plateaued. Taken together, those findings suggested that the genetic diversity of H6N6 viruses was far greater than that of H6N2 viruses during the period. Even though gene exchange in viruses with H6 lineage occasionally occurred, it remained rare or without endemic prevalence during that period, possibly due to transient reassortment viruses that decrease the fitness of the virus or due to established internal gene complexes that may benefit fitness [4]. As such, H6 subtype viruses differs greatly from other HPAIV, such as H5N8 and H7N9, which are highly adapted and spread stably in a relatively short period [42,43]. Our genetic analysis showed that most of the H6N1, H6N2, and H6N6 viruses were primarily isolated from Guangdong, Jiangxi, Guangxi, and Fujian Provinces (Figure 4), most likely due to the large population of waterfowl in southern China, which were regarded to be dominant reservoir of influenza viruses [4,40].

Phylogenetic analyses showed that H6 subtype viruses were primarily derived from Group I (e.g., ST339-like viruses), Group II (e.g., ST2853-like viruses), and Group III (e.g., HN573-like viruses), following the classification of past studies [4,35]. The surface genes of H6N2 viruses in our study clustered together and belonged to the Group I lineage, which is genetically different from the lineage of TW/2/13 human-like viruses. Nevertheless, the internal genes of the H6N2 viruses had clearly undergone complex reassortment. Meanwhile, the PB1 and the NP genes of H6N2 viruses clustered together and belonged to the Group III lineage. Although the NA and the NS genes of H6N2 viruses were classified in Group I, H6N2 viruses in this study had completely bifurcated into two evolutionary pathways. All those results implied that the H6N2 virus isolated from Guangdong Province had been undergoing multiple reassortment with different H6 lineages viruses. Frequent reassortment of AIVs is common in aquatic birds since the aquatic birds were regarded as the reservoirs of influenza viruses.

Despite past findings suggesting that interspecies transmissions of H6 subtype viruses from domestic ducks to terrestrial poultry are uncommon [4], our BSSVS procedure revealed that those viruses can be transmitted from aquatic birds to geese as well as from domestic ducks to chickens (Figure 6). Aquatic birds, including ducks and geese, are the main reservoir of AIVs, which can make them intermediate hosts between the real gene pool of migrating ducks and terrestrial poultry in the entire influenza virus ecosystem [4].

Among other findings, the transmission routes of H6 subtype viruses were primarily concentrated in the Jiangxi, Hunan, and Hubei Provinces of China, which suggested that those regions had hosted the most intensive and frequent transmission of the viruses, with Hubei emerging as a particularly interactive site. Studies have demonstrated the role of wild birds in the dispersal of influenza viruses during seasonal migration [44,45,46,47]. In the case of Hubei Province, the site of Poyang Lake, with its excellent ecology, vast wetlands, and luxuriant aquatic plants, has become a world-famous migratory bird migration and wintering hub. Because the production cycle of domestic ducks is synchronized with the migration of thousands of migratory birds in the area, viruses are frequently transmitted between waterfowl, which raised the risk of waterfowl-origin AIV transmitted to domestic poultry [48,49]. The spread of AIV is determined by interdependent factors, including wild bird migration, climate changes, and live poultry trade. Since waterfowl is the reservoirs of AIV, the broad prevalence of H6 subtype viruses from waterfowl accelerated the reassortment of different subtype AIVs. However, the correlation between the viral transmission and climate changes and the distribution of waterfowl in China is still unclear in this study, which requires further investigation. In light of that dynamic, more clearly understanding the spread of H6 subtype viruses in natural hosts is essential to reducing the risk of a viral spread in poultry and, in turn, mammalian hosts such as humans [47].

In our study, eight residues in the HA protein of H6 viruses exhibited positive selection. Among them, residue 174 surfaced in antigenic site B, while residues 145 and 156 were located at the receptor-binding region of the 130-loop. Our analysis indicated that those amino acid changes were positive selective sites, thereby suggesting that H6 subtype viruses were likely to undergo selection pressures. However, because such conclusions suffer from limitations in virus collection and functional verification, further investigation and surveillance are needed to better understand the ecology and the evolution of H6 subtype viruses.

Several types of vaccines have been developed against HPAIV subtypes, including H5 and H7 [50,51,52]. The most common subtypes in poultry—H6 and H9N2—have formed stable lineages, and both the H6 and H9 strains have been identified as providers of internal protein genes for the extensive distribution of the HPAIV H5N1 [53]. However, compared with other HPAIVs, H6 subtype viruses primarily infect waterfowl. Thus, if the government and the public does not pay sufficient attention and resources to the prevention and the control of viruses, it will continuously provide a gene pool for other HPAIV subtypes to exchange fragments, thereby resulting in the enhanced pathogenicity and transmission capacity of the reassortant virus [34]. Although vaccine immunization is not a panacea, H5 and H7 subtype influenza viruses can still undergo antigenic mutations after using vaccines in China [54]. Therefore, strengthening biological safety prevention and control is also an indispensable measure. For that reason, the evolutionary dynamics and the dissemination of H6 subtype influenza viruses in poultry demand immediate attention, especially to develop vaccines and to strengthen biosafety prevention able to prevent potential influenza pandemics in the future.

## Figures and Tables

**Figure 1 viruses-14-01154-f001:**
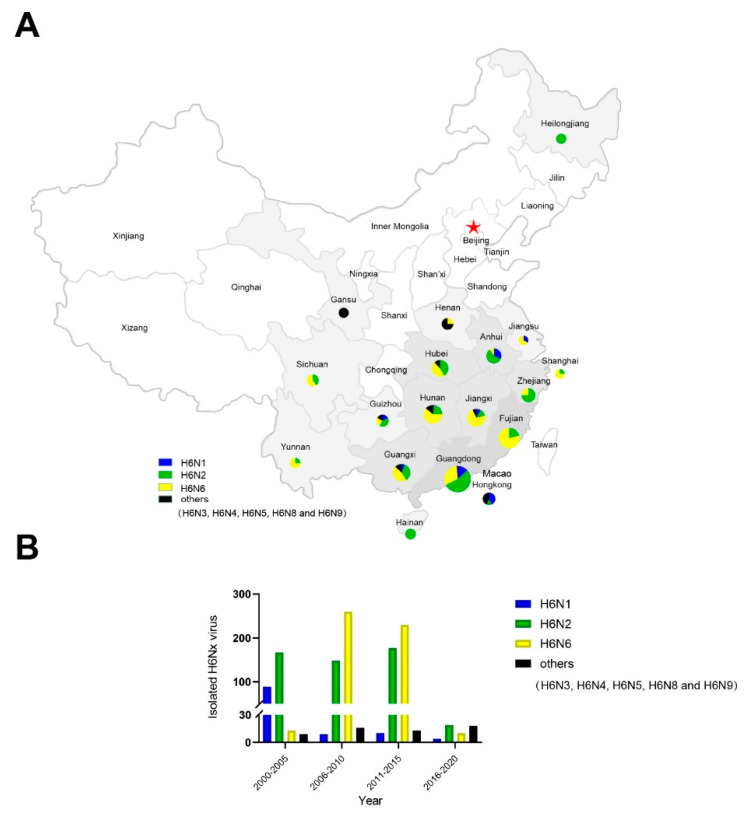
Geographic distribution of H6 subtype influenza viruses in China. (**A**) Distribution of H6Nx influenza viruses in China. In the pie charts, the size represents the number of H6 subtype influenza viruses and the colors represents H6 subtype viruses embodying different neuraminidases. The different background between provinces is indicated by the gray background color as areas isolated to the H6 subtype virus, while the white color indicates none. (**B**) The isolation rate of H6Nx influenza viruses isolated from China. Data are available from the GISAID’s EpiFlu Database and GenBank Database. The map was designed using ArcGIS Desktop 10.4 software.

**Figure 2 viruses-14-01154-f002:**
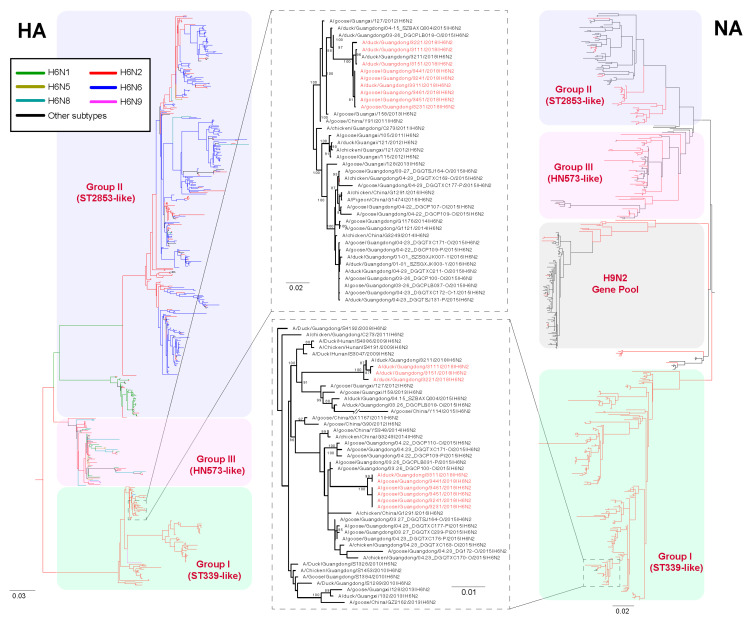
Maximum likelihood tree of HA and NA gene sequences of H6N2 viruses. All of the H6Nx viruses in China, including our new 10 H6N2 viruses, were used to perform the phylogenic analysis. All branch lengths are scaled according to the numbers of substitutions per site (subs/site). Maximum likelihood (ML) phylogenies for the codon alignment of the full gene segments were estimated using the GTR+G nucleotide substitution model in the RAxML (version 8.2) program. Node support was determined by nonparametric bootstrapping with 1000 replicates. The phylogenetic tree was visualized in the FigTree (version 1.4.3) program. Abbreviations: ST339, A/duck/Shantou/339/2000 (H6N2); ST2853, A/wild duck/Shantou/2853/2003 (H6N2); HN573, A/duck/Hunan/573/2002 (H6N2).

**Figure 3 viruses-14-01154-f003:**
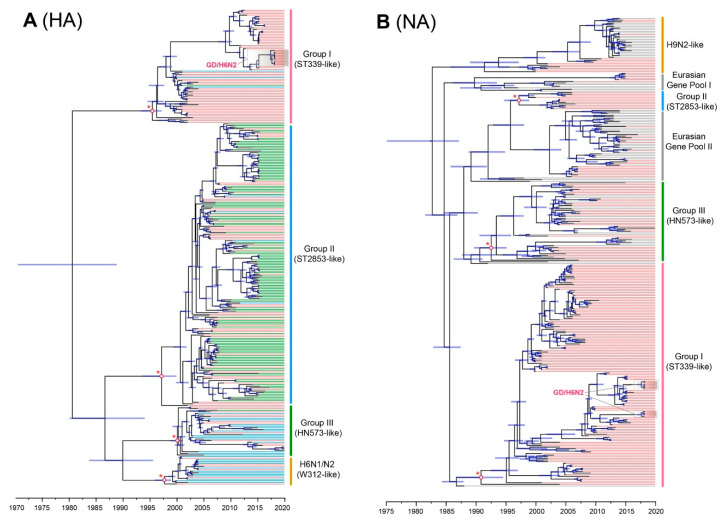
Time-scaled evolution of HA and NA genes of H6Nx viruses. (**A**) A maximum clade credibility tree of the HA sequence of H6Nx viruses sampled in China together with new H6N2 viruses is shown. H6Nx viruses containing N1, N2, and N6 subtypes are denoted by blue, red, and green colors, respectively. Shaded bars represent the 95% highest probability distribution for the age of each node. (**B**) A maximum clade credibility tree of the NA sequence of H6N2 viruses sampled in China together with new H6N2 viruses is shown. The phylogenetic tree was visualized in the FigTree (version 1.4.3) program. The H6N2 and the H9N2 subtypes are denoted by red and grey, respectively. Abbreviations: ST339, A/duck/Shantou/339/2000 (H6N2); ST2853, A/wild duck/Shantou/2853/2003 (H6N2); HN573, A/duck/Hunan/573/2002 (H6N2).

**Figure 4 viruses-14-01154-f004:**
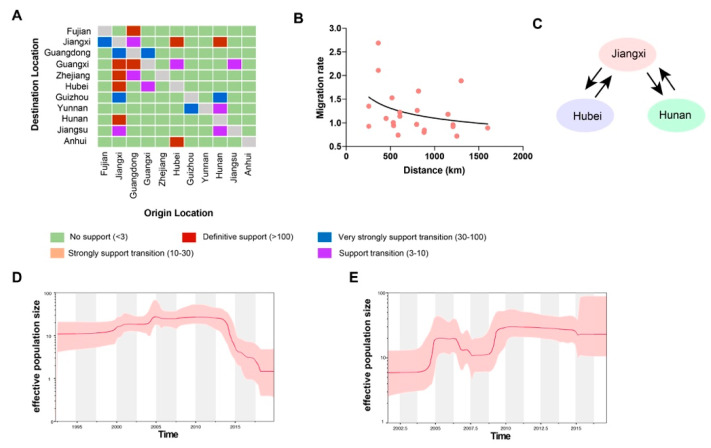
Evolutionary dynamics of H6Nx influenza viruses and level of Bayes factor support for each transmission route. (**A**) The transmission of H6Nx influenza viruses. The left and the right panels display the level of Bayes Factor (BF) support for each of the transmission routes considered for H6Nx influenza viruses. The *x*-axis represents the origin location and the *y*-axis represents the destination. (**B**) Among-province virus lineage transition rates (supported with Bayes factor >3) decrease with geodesic distance between provinces for H6Nx influenza viruses. (**C**) The transmission routes of H6 subtype viruses in Jiangxi, Hunan, and Hubei provinces. Bayesian Skyline plot of HA genes of prevailing H6N2 and H6N6 influenza viruses in China. A Bayesian Skyline analysis of HA gene of H6N2 viruses (**D**) and H6N6 viruses (**E**) to display changes in the effective population size over time. The solid red line indicates the median value and the shaded red area represents the 95% highest posterior density of genetic diversity estimates.

**Figure 5 viruses-14-01154-f005:**
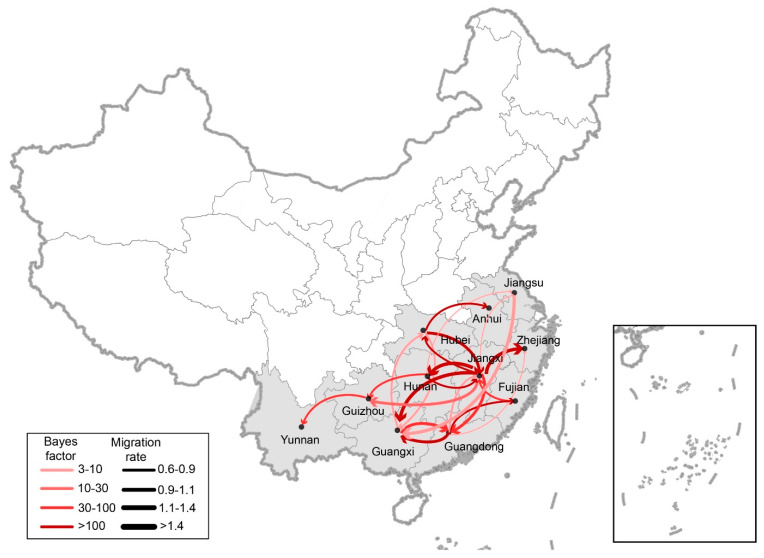
Spatiotemporal dissemination of H6Nx influenza viruses in China, which was determined by Bayesian phylogeographic inference of HA gene sequences. Curves show the among-province virus lineage transitions statistically supported with Bayes factor >3 for H6Nx influenza viruses. Curve widths represent transition rate values; curve colors represent corresponding statistical support (Bayes factor value) for each transition rate.

**Figure 6 viruses-14-01154-f006:**
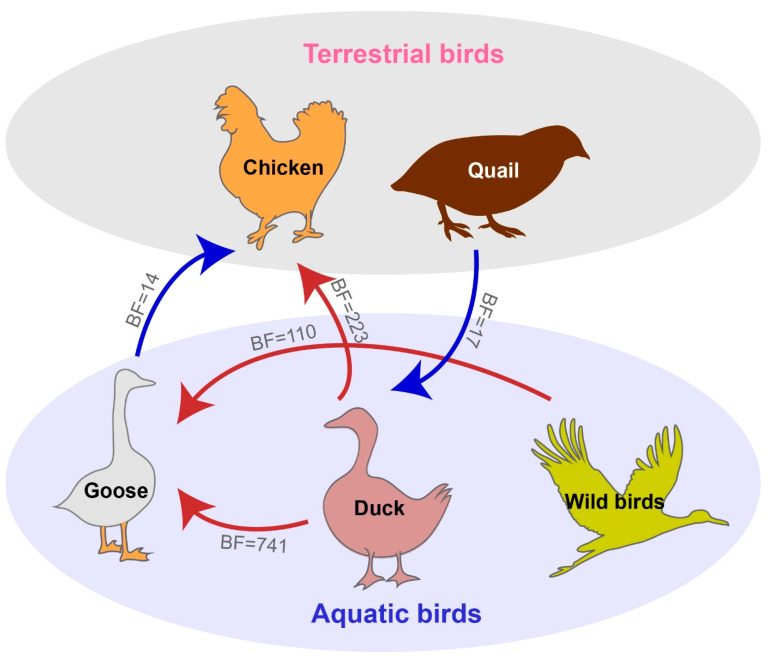
Host transition of H6Nx influenza viruses in China. Analyzing transition routes with Bayes factor (BF) values exceeding 10 were selected for analysis. The blue and red arrowhead represent the 10 < BF < 100 and BF > 100, respectively.

**Figure 7 viruses-14-01154-f007:**
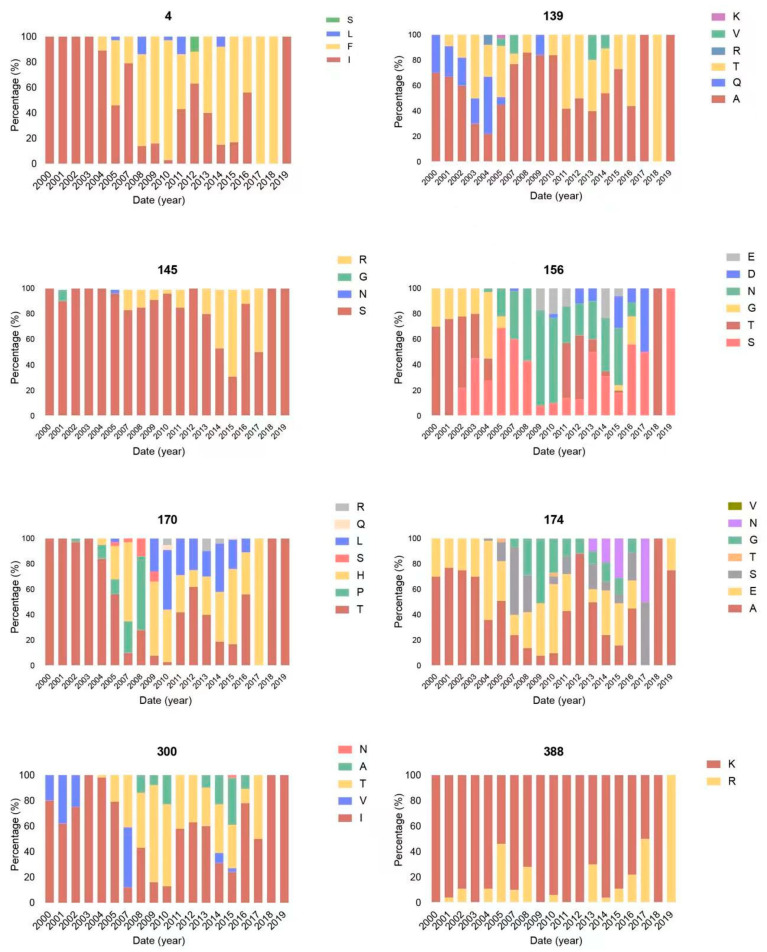
The positively selected sites of the HA protein of H6Nx viruses.

**Table 1 viruses-14-01154-t001:** Statistically supported migration rates of H6Nx influenza viruses in China estimated from HA gene sequences.

From	To	Bayes Factor	Posterior Probability	Migration Rate
Guangdong	Jiangxi	6	0.56	0.97
Guangxi	Hubei	6	0.55	0.96
Hubei	Guangxi	6	0.59	0.92
Jiangsu	Guangxi	6	0.57	0.89
Jiangxi	Jiangsu	7	0.52	0.74
Guangdong	Zhejiang	9	0.58	0.72
Hunan	Jiangsu	9	0.57	0.8
Jiangxi	Guizhou	16	0.61	1.18
Guizhou	Yunnan	30	0.75	1
Guangxi	Guangdong	31	0.75	1.14
Hunan	Guizhou	32	0.75	1.67
Jiangxi	Guangdong	65	0.86	1.26
Fujian	Jiangxi	69	0.87	0.94
Jiangxi	Guangxi	137	0.93	1.89
Jiangxi	Hubei	142	0.93	1.35
Jiangxi	Hunan	283	0.96	2.11
Hubei	Jiangxi	669	0.98	0.93
Jiangxi	Zhejiang	1348	0.99	1.53
Hubei	Anhui	1736	0.99	1.09
Hunan	Jiangxi	6510	1	2.69
Guangdong	Guangxi	24,441	1	1.22
Guangdong	Fujian	48,892	1	0.84

**Table 2 viruses-14-01154-t002:** Molecular characterization of H6Nx influenza viruses in China. DK, duck; GS, goose; SW, swine; TW, Taiwan; GD, Guangdong.

Virus Name	Amino Acid Sequence at Cleavage Site	Receptor-Binding Sites in HA						Key Position in HA			Amino Acid Deletion in NA (Position)	M2	PB2			NS1	PA
		137	138	169–171	186	190	226–228	156	263	380		31	588	627	701	92	38
DK/GD/3231/18(H6N2)	PQIETR↓ GLF	S	A	NNT	P	E	QKG	H	R	N	No	S	A	E	D	D	I
DK/GD/3111/18(H6N2)	PQIETR↓ GLF	S	A	NNT	P	E	QKG	H	R	N	No	S	A	E	D	D	I
GS/GD/3451/18(H6N2)	PQIETR↓ GLF	S	A	NNT	P	E	QKG	H	R	N	No	S	A	E	D	D	I
GS/GD/3441/18(H6N2)	PQIETR↓ GLF	S	A	NNT	P	E	QKG	H	R	N	No	S	A	E	D	D	I
DK/GD/3311/18(H6N2)	PQIETR↓ GLF	S	A	NNT	P	E	QKG	H	R	N	No	S	A	E	D	D	I
GS/GD/3241/18(H6N2)	PQIETR↓ GLF	S	A	NNT	P	E	QKG	H	R	N	No	S	A	E	D	D	I
GS/GD/3452/18(H6N2)	PQIETR↓ GLF	S	A	NNT	P	E	QKG	H	R	N	No	S	A	E	D	D	I
DK/GD/3151/18(H6N2)	PQIETR↓ GLF	S	A	NNT	P	E	QKG	H	R	N	No	S	A	E	D	D	I
GS/GD/3221/18(H6N2)	PQIETR↓ GLF	S	A	NNT	P	E	QRG	H	R	N	No	S	A	E	D	D	I
SW/GD/K6/10(H6N6)	PQIETR↓ GLF	N	S	RNT	P	A	QRS	H	S	N	No	S	A	E	D	D	I
TW/2/13(H6N1)	PQIATR↓ GIF	S	A	NNT	P	A	QRS	H	K	N	Yes (60–68)	S	A	E	D	D	I

## Data Availability

All the data needed to generate the conclusions made in the article are present in the article itself and/or the Appendix A. Additional data related to this article may be requested from authors.

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
