# Peer review of "The Genomic Evolution and the Transmission Dynamics of H6N2 Avian Influenza A Viruses in Southern China"

_viruses, 2022, doi:10.3390/v14061154_

Round 1
Reviewer 1 Report
In this manuscript, the authors analyzed the genomic evolution and transmission dynamics of H6 subtype influenza viruses in the southern parts of China. The background information about H6 subtype influenza viruses and experimental procedures were well described, and the data were well presented, interpreted, and discussed. One of the major findings was that most of the H6N1, H6N2, and H6N6 viruses were found to be in Guangdong, Jiangxi, Guangxi and Fujian provinces. Because there are large populations of waterfowl in these provinces, the authors postulated that waterfowl is likely to be a major factor that affects the distribution of these subtype viruses. Can the authors provide a map showing the waterfowl distribution in China to see if the distribution of the viruses and the distribution of waterfowl overlap? Can other factors affect the distribution of these subtype viruses in China, for example, temperature (as temperature in these provinces are high)? Another major issue is that the authors need to pay more attention to the language. There are many obvious spelling and grammar errors.
Author Response
Response: Thanks for the reviewer’s suggestion. We fully agree with the reviewer’s viewpoint that waterfowl is likely to be a major factor that affects the distribution of these subtype viruses, and it is helpful to see the distribution of waterfowl in China and temperature changes; however, due to the limitation to access these data and difficult to conducted in the GLM analysis, we have added discussion and limitation about the phenomenon in the discussion section. It is an important scientific problem, and we will focus on this analysis in the future.“The spread of AIV is determined by interdependent factors, including wild bird migration, climate changes, and live poultry trade. Since waterfowl is the reservoirs of AIV, the broad prevalence of H6 subtype viruses from waterfowl accelerated the reassortment of different subtype AIVs. However, the correlation between the viral transmission and climate changes and the distribution of waterfowl in China is still unclear in this study, which requires further investigation.” In addition, we also checked the grammar errors carefully in the full manuscript. Please see page 16 lines 430-433; page 17 lines 451-457.

Reviewer 2 Report
This study provided recent molecular epidemiological information of H6-subtype AIVs found in southern China. The authors analyzed 10 H6N2 AIVs with references and showed phylogenetic relationship, Evolutionary dynamics, Bayesian phylogeographic analysis, host transition analysis, a.a analysis, and positive selection site analysis results. A part of epidemiological interpretations was well written, but some part of host switching, and positive selection sites should be more discussed. Especially, additional discussion on the positive selection sites and their potential characteristics with other related papers will improve the manuscript for the readers in the related field. Other points are also listed as below:
- Line 44-47: Sometimes HPAIV infection could be fatal in wild aquatic bird. In this regard, the contents about ‘no clinical symptoms in aquatic bird infected with AIVs’ should be modified.
Kleyheeg E, Slaterus R, Bodewes R, et al. Deaths among Wild Birds during Highly Pathogenic Avian Influenza A(H5N8) Virus Outbreak, the Netherlands. Emerg Infect Dis. 2017;23(12):2050-2054.
- Line 71: mammalian mutations -> mammalian species ?
- Line 97: How many samples were tested? In addition, Ethic statement for the animal sampling is recommended.
- Line 105: Table s1 which described 10 H6N2 AIV sequences should be cited somewhere of the paragraph.
- Line 116, 118, 124: Please put references for each analysis tool. Other tools were also put with references.
- Line 118: GTRGAMMA seemed to be a typo. Please put the exact substitution model.
- Line 136-138: Did the authors have any reason to select the models described here?
- Line 191: repeated phrase “suggested that”.
- Line 211: Please put a reference about the previous study.
- Line 213-222: A table regarding the segment genomes of H6N2 isolates will be helpful for the readers to understand the explanation.
- Line 260: (B) Please indicate what the color differences mean.
- Table 2 should be reconstructed. Current form is in a little bit of torsion.
Author Response
- Line 44-47: Sometimes HPAIV infection could be fatal in wild aquatic bird. In this regard, the contents about ‘no clinical symptoms in aquatic bird infected with AIVs’ should be modified.
Kleyheeg E, Slaterus R, Bodewes R, et al. Deaths among Wild Birds during Highly Pathogenic Avian Influenza A(H5N8) Virus Outbreak, the Netherlands. Emerg Infect Dis. 2017;23(12):2050-2054.
Response: Author agree with the reviewer. We have modified the sentence. “Aquatic birds are major reservoirs of AIVs, which spread rapidly among wild birds”. Please see page 2 lines 43-45.
- Line 71: mammalian mutations -> mammalian species?
Response: Author agree with the reviewer. We have modified the words to “mammalian species”. Please see page 2 lines 73-74.
- Line 97: How many samples were tested? In addition, Ethic statement for the animal sampling is recommended.
Response: Author agree with the reviewer. We have added the sampling number and the Ethic statement for the animal sampling. “From January to December 2018, we collected 1,348 cloacal and tracheal swab samples of chickens, ducks, and geese from live poultry markets in Guangdong, China. All animals involved in experiments were reviewed and approved by the Institution Animal Care and Use Committee at South China Agricultural University and treated in accordance with the guidelines (2017A002).” Please see page 3 lines 101-105.
- Line 105: Table s1 which described 10 H6N2 AIV sequences should be cited somewhere of the paragraph.
Response: Author agree with the reviewer. We have added the sentence in the manuscript.
The detailed information of gene sequences was available from Supplementary Table 1.Please see page 3 lines 119-122.
- Line 116, 118, 124: Please put references for each analysis tool. Other tools were also put with references.
Response: Author agree with the reviewer. We have added the references. Please see page 3 lines 130,132; page 4 lines139,162, 183.
- Line 118: GTRGAMMA seemed to be a typo. Please put the exact substitution model.
Response: Author agree with the reviewer. We have revised the word to “GTR+G”. Please see page 3 line 132.
- Line 136-138: Did the authors have any reason to select the models described here?
Response: Because multiple papers showed that the model is common in the Bayesian phylogenetic analysis of influenza virus, we choose the model to conduct the analysis. In addition, we have also conducted model selection and found that this model is much suitable for the analysis.
- Line 191: repeated phrase “suggested that”.
Response: Author agree with the reviewer. We have deleted it. Please see page 5 line 208.
- Line 211: Please put a reference about the previous study.
Response: Author agree with the reviewer. We have added the references of the previous study. Please see page 6 line 227.
- Line 213-222: A table regarding the segment genomes of H6N2 isolates will be helpful for the readers to understand the explanation.
Response: Author agree with the reviewer. We have added the table regarding the segment genomes of H6N2 isolates. Please see Supplementary Table S2.
- Line 260: (B) Please indicate what the color differences mean.
Response: Author agree with the reviewer. We have added the meaning of the color. Please see page 8 lines 286-292.
- Table 2 should be reconstructed. Current form is in a little bit of torsion.
Response: Author agree with the reviewer. We have reconstructed the Table 2.

Reviewer 3 Report
The manuscript “Genomic evolution and transmission dynamics of H6N2 avian influenza A viruses in southern China” by Zhou and co-authors performed genetic evolutionary and transmission dynamic analyses of avian influenza viruses of H6 subtype and mainly H6N2. The analyses were based on the whole genome sequence of data retrieved from the public data base as well as additional few collected samples isolated from birds in China. The presented study offers interesting data and is well written. However, the following points should be considered and addressed by the authors in the manuscript prior to being submitted for publication as detailed below:
-Line 62: “H6 subtype viruses, have increasingly burdened the poultry industry since first being isolated in a turkey in Massachusetts in 1965” The H6 is mainly increased in the last decade.
-Line 76: is there any human cases with H6N2?
-Line 98: how many samples were collected during the eight-month period? And please specify for each species.
-Line 105: “RNA was extracted from the suspension of 10 H6N2 influenza viruses” do the authors mean samples or positive control?
-Line 106: did the authors perform Matrix RT-qPCR to screen for general influenza? Or tested the collected samples against other avian influenza subtypes?
-Line 108: what are the sequencing conditions/kit used for sequence reaction and sequence product purification?
-Line 112: “All available full genome sequences with the complete coding regions of H6 subtype” please specify the date when you retrieved your data and the total number of each gene segment?
-Line 117: what about the model used for other genes?
-Line 138: “Multiple runs” how many? Please specifiy?
-Line 152: “We accessed all available HA genome sequences of H6 subtype viruses circulating in China from the GISAID Database.” Again please specify the date in which you collected the sequence and total number?
-Line162: why the substitution model here (GTR) is not the same as line 136 which is SRD06
-Line 189: results obtained from collected samples should be described at the beginning of the results section
-Line 195-198: please mention numbers or percentage of detected cases.
-Line 259: “other subtypes” please specify!
-Lines 345-350 and discussion: can the author describe more about the role AA under pressure and what does this mean?
-Minor English check is required: for example: line 178, line 190, line 197, line 198 ….etc..
Author Response
-Line 62: “H6 subtype viruses, have increasingly burdened the poultry industry since first being isolated in a turkey in Massachusetts in 1965” The H6 is mainly increased in the last decade.
Response: Author agree with the reviewer. We have revised the sentence. “H6 subtype viruses, was first isolated in a turkey in Massachusetts in 1965, and the H6 subtype virus had become prevalent worldwide over the past decade, causing huge losses to the poultry industry burden”. Please see page 2 lines 61-65.
-Line 76: is there any human cases with H6N2?
Response: There is no human cases with H6N2, and only H6N1 viruses were reported infected humans.
-Line 98: how many samples were collected during the eight-month period? And please specify for each species.
Response: We have added the species and sampling numbers in the manuscript. “From January to December 2018, we collected 1,348 cloacal and tracheal swab samples of chickens, ducks, and geese from live poultry markets in Guangdong, China. From January to December 2018, we collected 1,348 cloacal and tracheal swab samples of chickens, ducks, and geese from live poultry markets in Guangdong, China.” Please see page 3 lines 101-105.
-Line 105: “RNA was extracted from the suspension of 10 H6N2 influenza viruses” do the authors mean samples or positive control?
Response: The authors means the samples. We have modified the sentence to make it more clear. “Before amplication of the full-length genome sequences, virus isolation from the swab samples were conducted, and the HA and NA genes of positive samples were firstly amplified and identified. Then, RNA was extracted from the suspension of 10 H6N2 influenza viruses with the RNeasy Mini Kit (QIAGEN) as directed by the manufacturer.” Please see page 3 lines 111-122.
-Line 106: did the authors perform Matrix RT-qPCR to screen for general influenza? Or tested the collected samples against other avian influenza subtypes?
Response: We have added more details about the surveillance information. “Before amplication of the full-length genome sequences, virus isolation from the swab samples were conducted, and the HA and NA genes of positive samples were firstly amplified and identified. Then, RNA was extracted from the suspension of 10 H6N2 influenza viruses with the RNeasy Mini Kit (QIAGEN) as directed by the manufacturer. Two-step RT-PCR was conducted with universal primers as previously described [24]. Eight gene sequences of H6N2 viruses were amplified using PrimeSTAR Max DNA Polymerase (TAKARA) with frame-specific primers. PCR products were purified with a Gel Extraction Kit D2500 (OMEGA), and the gene sequences were sequenced by TSINGKE Co.,Ltd. (Guangdong, China).” Please see page 3 lines 111-122.
-Line 108: what are the sequencing conditions/kit used for sequence reaction and sequence product purification?
Response: We have added the sequencing conditions/kit used for sequence reaction and sequence product purification. “Eight gene sequences of H6N2 viruses were amplified using PrimeSTAR Max DNA Polymerase (TAKARA) with frame-specific primers. PCR products were purified with a Gel Extraction Kit D2500 (OMEGA), and the gene sequences were sequenced by TSINGKE Co.,Ltd. (Guangdong, China).” Please see page 3 lines 111-122.
-Line 112: “All available full genome sequences with the complete coding regions of H6 subtype” please specify the date when you retrieved your data and the total number of each gene segment?
Response: We have added more information about the total number of each gene segment and the date we retrieved the data. “The number of reference sequences of PB2, PB1, PA, HA, NP, NA, M, and NS were 568, 527, 558, 532, 570, 411, 499, and 425, respectively (accessed 18 September 2020).” Please see page 3 lines 126-128.
-Line 117: what about the model used for other genes?
Response: We have added the model used for other genes. “Maximum likelihood (ML) phylogenies for the codon alignment of eight gene segments were estimated using the GTR+G nucleotide substitution model in RAxML (version 8.2) program.” Please see page 3 line 131.
-Line 138: “Multiple runs” how many? Please specifiy?
Response: Author agree with the reviewer. It means two runs. We have modified it. Please see page 4 line 152.
-Line 152: “We accessed all available HA genome sequences of H6 subtype viruses circulating in China from the GISAID Database.” Again please specify the date in which you collected the sequence and total number?
Response: We accessed 868 HA genome sequences of H6 subtype viruses circulating in China from the GISAID Database. Please see page 4 line 166.
-Line162: why the substitution model here (GTR) is not the same as line 136 which is SRD06
Response: We are sorry for the error. The correct model is GTR. We have revised it. Please see page 4 line 150.
-Line 189: results obtained from collected samples should be described at the beginning of the results section
Response: We have added the results obtained from collected samples at the beginning of the results section. “From January to December 2018, we collected 1,348 cloacal and tracheal swab samples of chickens, ducks, and geese from live poultry markets in Guangdong, China. A total of 10 H6N2 viruses were isolated during this period”. Please see page 3 lines 102-105.
-Line 195-198: please mention numbers or percentage of detected cases.
Response: In this section, we mainly focus on the increasing trends of the number of H6 subtype viruses. For the H6 subtype viruses cover at least 18 provinces, it is difficult to describe the percentage of different subtype viruses in different provinces.
-Line 259: “other subtypes” please specify!
Response: We have specified the other subtypes. “H6Nx viruses containing N1, N2, and N6 subtypes are denoted by blue, red, and green colors, respectively.” Please see page 8 lines 286-287.
-Lines 345-350 and discussion: can the author describe more about the role AA under pressure and what does this mean?
Response: Author agree with the reviewer. We have checked the papers and found that no biological characterizations of these amino acids were reported; therefore, we have discussed that the function of these amino acid substitutions requires further investigation. We will focus on these amino acid substitutions in the future.
-Minor English check is required: for example: line 178, line 190, line 197, line 198 ….etc..
Response: Author agree with the reviewer. We have checked them carefully.
